# GANet: Graph-Aware Network for Point Cloud Completion with Displacement-Aware Point Augmentor

## Abstract

Remarkably, real-world data (e.g., LiDAR-based point clouds) is commonly sparse, uneven, occluded, and truncated. The point cloud completion task draws due attention, which aims to predict a complete and accurate shape from its partial observation. However, existing methods commonly adopt PointNet or PointNet++ to extract features of incomplete point clouds. In this paper, we propose an end-to-end Graph-Aware Network (**GANet**) to effectively learn from the contour information of the partial point clouds. Moreover, we design Displacements-Aware Augmentor (DPA) to upsample and refine coarse point clouds. With our graph-based feature extractors and Displacements-Aware Transformer, our DPA can precisely capture the geometric and structural features to refine the complete point clouds. Experiments on PCN and MVP datasets demonstrate that our GANet achieves state-of-the-art on the task of shape completion.

## 1 Introduction

The rapid development of 3D scanning devices (e.g.. LiDAR) has provided an unprecedented ability to capture point clouds from complex 3D scenes. However, due to limited resolution and occlusion issues, the scanned point clouds are sparse and incomplete, which is why various applications such as 3D detection (Zhang et al., 2020b) cannot take full advantage of them.

PointNet (Qi et al., 2017a) has attracted great attention to the learning methods on raw point clouds. Inspired by PointNet, PCN (Yuan et al., 2018) introduces a coarse-to-fine fashion to learning-based shape completion. Based on this fashion, subsequent work (Yuan et al., 2018; Tchapmi et al., 2019; Wang et al., 2020a; Liu et al., 2020; Wang et al., 2020b; Pan et al., 2021) investigates how to optimize the refinement stage for more detailed results. For example, SnowflakeNet (Xiang et al., 2021a) proposes snowflake point deconvolution to progressively refine coarse point clouds.

Although difficult to discern as a whole, most incomplete point clouds maintain roughly recognizable contours. This observation motivates us to propose Graph-Aware Network (**GANet**), a novel graph-based network for shape completion. Compared with MLP-based methods previous approaches, which rely heavily on inductive learning and may neglect shape awareness as mentioned in Liu et al. (2019b), graph-based methods can extract shape information from the hints of geometric relation more effectively. An overview of our GANet is shown in Figure 1. Specifically, we design a Multi-scale Edge Aggregator (**MEA**) to extract expressive features with rich geometric information. The MEA first applies a set abstraction proposed by PointNet++ (Qi et al., 2017b) to reduce the point number of input data. This operation avoids the effects of noise and repeated points as well as reduces the model's computation complexity. To learn from the outlines of the partial point clouds, we construct the local graphs based on the neighbors of the given centroids. Then we propose a novel scalable module, Local Edge Aggregator (LEA) to process the local graphs. This module weights the importance of the edges in the local graphs and then aggregates the features of the edges for the output centroid features. In addition, to capture both the local and global structures of the input point clouds, we introduce the philosophy of multi-scaling to our LEA.

Furthermore, we design Displacement-Aware Point Augmentor (**DPA**), a novel upsampling module to refine the coarse output. We leverage a multi-stage strategy to stack DPA blocks. In particular, we use the LEA as the feature extractor in every DPA block. The LEA can capture the geometric

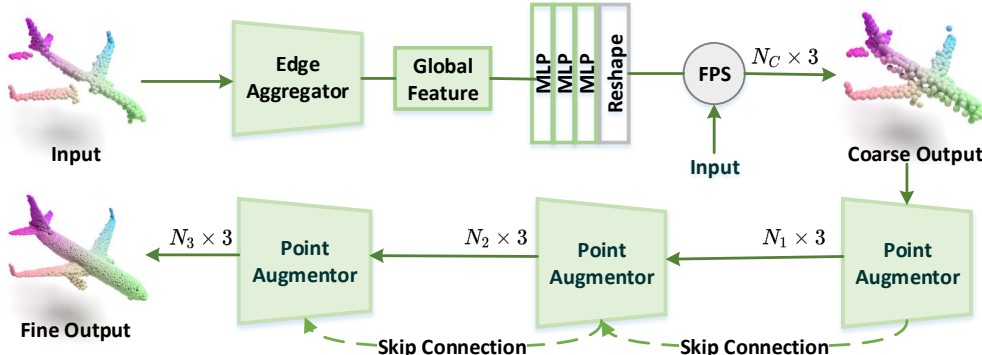

Figure 1: The pipeline of our GANet. GANet consists of an edge aggregator (our MEA) to extract the global features of a partial point cloud, and MLP to decode the global features for the coarse output, a priori sampling module for stable output, and three cascaded point augmentors (our DPA) with skip connection to upsample and refine the coarse output. $N_c$ is the number of the coarse output points. $N_1$, $N_2$, and $N_3$ are the number of the output points from the three point augmentors.

structure information to provide better representation compared with regular multi-layer perceptron (MLP). In addition, the current DPA block provides information flow for the next DPA block. This skip connection can help the next block to refine a better complete point cloud with information fusion. To effectively utilize previous information, we propose a novel Displacements-Aware Transformer (DAFormer). With the cross attention and group feed-forward network, our DAFormer can learn the precise displacement relation between two point clouds with different resolutions. Finally, our experiments show that GANet outperforms previous methods on PCN dataset (Yuan et al., 2018) and MVP (Pan et al., 2021).

Our key contributions are manifold:

- We design a new Graph-Aware Network (GANet) for point cloud completion, which uses the graph-based scalable module to extract local and geometric features of partial point cloud.
- We propose a Displacements-Aware Point Augmentor (DAP) to refine coarse complete point clouds. Moreover, we introduce a novel Displacements-Aware Transformer (DAFormer) to aggregate information between two point clouds with different resolution.
- Our method achieves state-of-the-art on some widely adopted benchmarks including PCN dataset and MVP.

## 2 RELATED WORK

### 2.1 LEARNING ON POINT CLOUDS

**Point-based methods** (Li et al., 2018a; Zhao & Tao, 2020; Yang et al., 2019; Liu et al., 2019a; Zhao et al., 2021; Guo et al., 2021; Xiang et al., 2021b; Ma et al., 2021; Ran et al., 2022) have attracted significant attention for processing on point clouds. PointNet (Qi et al., 2017a) learns a global view by point-wise MLP followed by max-pooling. Subsequently, PointNet++ (Qi et al., 2017b) introduces a hierarchical framework to learn the local features. Afterward, another branch of methods (Dai et al., 2017; Groh et al., 2018; Thomas et al., 2019; Li et al., 2018b; Xu et al., 2018; 2021) based on convolution emerged for the local aggregation, using dynamic strategies of transformation for the normal work of convolution on point clouds. PointConv (Wu et al., 2019) directly employs the relationship between local centers and their neighbors to learn a dynamic weight for convolution.

### 2.2 GRAPH-BASED METHODS

(Groueix et al., 2018; Xu et al., 2020a;b; Hamilton et al., 2017; Kipf & Welling, 2016; Zhou et al., 2021) achieves notable performance for the local aggregation of geometric features. To capture local

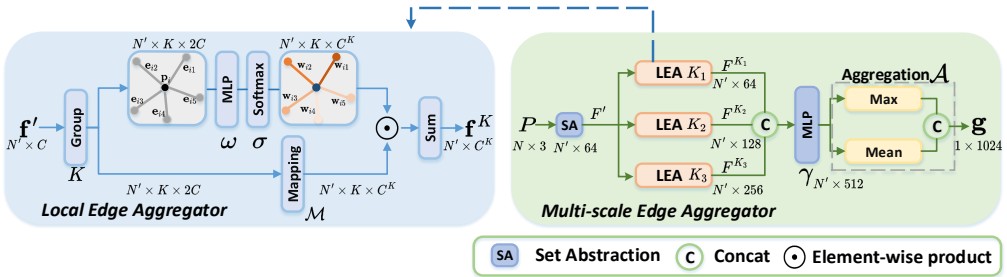

Figure 2: The design of our Local Edge Aggregator (left) and Multi-scale Edge Aggregator (right). Local Edge Aggregator (LEA) describes the local geometric structure $\mathbf{f}^K$ by a combination of graph aggregation and attention mechanism, and Multi-scale Edge Aggregator (MEA) adopts the philosophy of multi-scaling for more expressive features. We first utilize a set abstraction block (SA) to extract some key points $P'$ and the corresponding features $F'$. Then, we feed $F'$ into LEAs with different $K$ (i.e., $K_1$, $K_2$, $K_3$) and obtain the output $F^{K1}$, $F^{K2}$, $F^{K3}$. Finally, through a sequence of operations (i.e., concatenation, mapping $\gamma$, and aggregation $\mathcal{A}$), our MEA abstracts a local-to-global view of feature $\mathbf{g}$. $N$ is the number of input points. $N'$ is the number of key points after SA. $K$ is the number of neighbors to be queried. $C$ and $C^K$ are the dimensions of the features in $F'$ and $F^K$.

geometric features of point clouds, DGCNN (Wang et al., 2019) learns an edge feature by building the relations between the node and its neighbors. A similar branch of relation-based models learns from relations analogous to the edges of graphs. RSCNN (Liu et al., 2019c) predefines geometric priors equipped with the attention mechanism, while RPNet (Ran et al., 2021) adopts both the geometric and semantic relations in an efficient block.

## 2.3 LEARNING-BASED SHAPE COMPLETION

Various shape completion methods (Yang et al., 2018; Vakalopoulou et al., 2018; Tchapmi et al., 2019; Liu et al., 2020; Xie et al., 2020; Huang et al., 2020; Wang et al., 2020b; Zhang et al., 2020a) have emerged. PCN (Yuan et al., 2018) unprecedentedly introduces deep learning to shape completion by utilizing an coarse-to-fine completion framework. CRN (Wang et al., 2020a) proposes a novel coarse-to-fine pipeline to facilitate the decoder with a cascade refinement strategy. Subsequent work (Yu et al., 2021; Huang et al., 2021; Zong et al., 2021; Wen et al., 2021a;b; Lyu et al., 2021; Wang et al., 2021; 2022; Tang et al., 2022) focuses on how to reconstruct more detailed features. VRCNet (Pan et al., 2021) designs a Relational Enhancement Network to enhance structural relations for refinement. SnowflakeNet (Xiang et al., 2021a) introduces a snowflake point deconvolution block to generate the detailed complete point cloud.

## 3 GRAPH-AWARE NETWORK

In this section, we propose **G**raph-**A**ware **Net**work (**GANet**) for shape completion, following the coarse-to-fine fashion. As shown in Figure 1, GANet firstly extracts the global features by our Multi-scale Edge Aggregator (MEA) to predict the coarse output. Afterwards, we upsample the coarse output to fine complete point clouds through our Displacements-aware Point Augmentor modules. Finally, we present our Displacement-aware Transformer, loss function and the evaluation metrics.

### 3.1 MULTI-SCALE EDGE AGGREGATION

Prior PointNet-based methods (Yang et al., 2018; Vakalopoulou et al., 2018; Tchapmi et al., 2019; Liu et al., 2020; Yuan et al., 2018; Wang et al., 2020a; Pan et al., 2021) can hardly learn geometric information as they learn the global and local features from individual points. To tackle this problem, we exploit the geometric structures by constructing local graphs. We design a Local Edge Aggregator (LEA), an attention-based instead of the convolution-like module to aggregate the local graphs. ~~Furthermore, we use a different definition of edge.~~ Based on LEA, we propose a novel

Multi-Scale Edge Aggregation (MEA) to extract global features. As shown in Figure 2, the MEA consists of a set abstraction operation and three Local Edge Aggregators (LEA). The set abstraction is employed to extract key points and their features as well as drop the computation complexity.

As shown in Figure 2, we denote the input as $P = \{\mathbf{p}_1, \ldots, \mathbf{p}_N\} \subseteq \mathbb{R}^{N \times 3}$. $N$ is the number of the input point clouds. The output of set abstraction (SA) is a downsampled point cloud $P' = \{\mathbf{p}'_1 \ldots, \mathbf{p}'_{N'}\} \subseteq \mathbb{R}^{N' \times 3}$ and its corresponding features $F' = \{\mathbf{f}'_1 \ldots, \mathbf{f}'_{N'}\} \subseteq \mathbb{R}^{N' \times C}$. $N'$ is the number of points in the downsampled point cloud, and $C$ means the dimension of features $F'$. We then feed $F'$ into the multi-scale LEAs. Given the $i$-th point, its output feature through one LEA block can be formulated as:

$$\mathbf{f}_i^K = \sum_{j=1}^{K} \mathbf{w}_{ij} \odot \mathcal{M}\left([\mathbf{f}'_i, \mathbf{e}_{ij}]\right), \tag{1}$$

where we define edge $\mathbf{e}_{ij} = \mathbf{f}'_i - \mathbf{f}'_j$, and $\mathbf{w}_{ij}$ is the attention weight of $\mathbf{e}_{ij}$. $K$ represents the number of neighbors to be queried by one LEA. $\odot$ is element-wise product. $[\cdot]$ is the operation of concatenation. $\mathcal{M}$ is a combination of linear and non-linearity function, i.e., $\{MLP \rightarrow ReLU \rightarrow MLP\}$. Here $\mathbf{w}_{ij}$ is defined as:

$$\mathbf{w}_{ij} = Softmax\left(\omega\left(\mathbf{e}_{ij}\right)\right), \tag{2}$$

where $\omega$ is a learnable mapping function.

Furthermore, we introduce the philosophy of multi-scale grouping to our edge aggregation. Previous works (Qi et al., 2017b; Ran et al., 2021) prove the effectiveness of multi-scale grouping. Both the local and global structures are essential for a sparse point cloud. Empirical results further show its effectiveness. The global feature of our multi-scale LEAs can be defined as:

$$\mathbf{g} = \mathcal{A}\left(\left\{\gamma\left(\left[\mathbf{f}_i^{K_1}, \mathbf{f}_i^{K_2}, \mathbf{f}_i^{K_3}\right]\right) | i \in \{1, \ldots, N'\}\right\}\right), \tag{3}$$

where $K_1$, $K_2$, $K_3$ and $\mathbf{f}_i^{K_1}$, $\mathbf{f}_i^{K_2}$, $\mathbf{f}_i^{K_3}$ are the predefined numbers of neighbors and the output of different scales of LEAs, respectively. $\gamma$ is a linear function. To implement our network, we set $K_1$, $K_2$, $K_3$ to 10, 20, None (all points as neighbors) respectively. $\mathcal{A}$ is the function (i.e., max-pooling, mean-pooling) to aggregate the $N'$ features.

Finally, we utilize the input partial point cloud and the extracted global feature $\mathbf{g}$ to generate a coarse complete point cloud through a sequence of operations, i.e., mapping, reshaping, and sampling.

## 3.2 DISPLACEMENT-AWARE POINT AUGMENTATION

The upsampling operation plays a vital role in the point cloud completion task. In this stage, the aim is to refine and upsample coarse point clouds. With a multi-stage strategy, previous methods (Wang et al., 2020a; Xiang et al., 2021a; Tang et al., 2022) can recover local shape details by exploiting local points features. However, these multi-stage methods ignore the importance of the feature extractor. They usually employ MLP-based feature extractor to learn the features of the input, which may fail to exploit the geometric and structural features of the input.

To solve the above problems, we propose a Displacement-aware Point Augmentation (DPA) block, as shown in Figure 3. Following a multi-stage strategy, we stack three DPA blocks to generate our final complete shape. Each DPA block takes the output of the previous block as input and then refines and upsamples the input with more details. The simple yet effective DPA block consists of a feature extractor, feature fusion, and displacement generator. We adopt one LEA as our feature extractor for the local feature extraction. This process is roughly the same as Equation 1.

In addition, to refine a better point cloud, we aggregate point features of $i-1$-th block to the current $i$-th block. To fuse two different scale features, we construct a relation between current features $F_i$ and previous point features $F_{i-1}$. The relation can guide the DPA block to generate suitable and accurate displacements by aggregating information from $P_{i-1}$ and $P_i$. To fuse two point features, we propose Displacement-aware Transformer (DAFormer).

Specifically, the input of DPA is point cloud $P_i = \left\{\mathbf{p}_1^i, \ldots, \mathbf{p}_{N_i}^i\right\} \subseteq \mathbb{R}^{N_i \times 3}$, where $N_i$ is its number of points. Our goal is to obtain a refined point cloud $P_{i+1} = \left\{\mathbf{p}_1^{i+1}, \ldots, \mathbf{p}_{N_{i+1}}^{i+1}\right\} \subseteq \mathbb{R}^{N_{i+1} \times 3}$. We

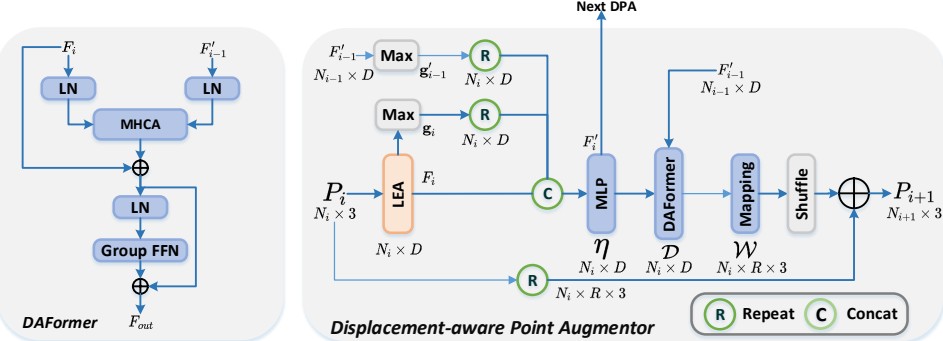

Figure 3: Displacement-aware Point Augmentator (DPA). Our DPA aims to generate a high-resolution fine output $P_{i+1}$ from the coarse input $P_i$. We first adopt a LEA to extract the local features $F_i$. Next, we feed $F_i$ with the combination of the feature after max-pooling $\mathbf{g}_i$ and the one from the previous stage $\mathbf{g}'_{i-1}$ into an MLP $\eta$ to obtain features $F'_i$. The pooled feature $\mathbf{g}'_i$ will be used in the next DPA block. Then, we leverage DAFormer $\mathcal{D}$ along with the mapping function $\mathcal{W}$ and the pixel-shuffle-like operation, to generate point displacements with the fusion of $F'_i$ and features from the previous stage $F'_{i-1}$. Finally, we use the computed point displacements to obtain the refined output $P_{i+1}$. $R$ is upsampling ratio. $N_i$ is the number of input points $P_i$. $N_{i+1}$ is the number of output points $P_{i+1}$. $N_{i+1} = R \times N_i$.

firstly adopt one LEA to extract local features $F_i = \left\{ \mathbf{f}_1^i, \ldots, \mathbf{f}_{N_i}^i \right\} \subseteq \mathbb{R}^{N_i \times D}$. Next, we feed $F_i$ with the combination of the feature after max-pooling $\mathbf{g}_i$ and the one from the previous stage $\mathbf{g}'_{i-1}$ into an MLP $\eta$ to obtain features $F'_i$. The pooled feature $\mathbf{g}'_i$ will be used in the next DPA block. Then, DAFormer $\mathcal{D}$ along with the mapping function $\mathcal{W}$ and the pixel-shuffle-like operation, utilizes the fusion of $F'_i$ and features from the previous stage $F'_{i-1}$ to generate point displacements. We define a set of point displacements $D_j$ as:

$$D_j = \left\{ \left( \mathcal{W} \left( \mathcal{D} \left( \mathbf{f}_j^{i-1}, \eta \left( \left[ \mathbf{f}_j^i, \mathbf{g}_{i-1}, \mathbf{g}_i \right] \right) \right) \right) \right)^m \mid m \in \{1, \ldots, R\} \right\}. \tag{4}$$

Thus, after point augmentation, each point of the input $P_i$ will generate $R$ corresponding points ($N_{i+1} = R \times N_i$).

Finally, we obtain the refined point cloud $P_{i+1}$ by the summation of the displacements and their corresponding centroids. Denote $\mathbf{d}_{jm}$ as an element of $D_j$. The refined point cloud can be defined as:

$$P_{i+1} = \left\{ \mathbf{p}_j^i + \mathbf{d}_{jm} \mid j \in \{1, \ldots, N_i\}, m \in \{1, \ldots, R\} \right\}. \tag{5}$$

### 3.3 DISPLACEMENT-AWARE TRANSFORMER

In the upsampling stage, we learn point displacements to refine the point cloud. To learn better point displacements, we introduce a novel Transformer, called Displacement-aware Transformer (DAFormer). As shown on the left of Figure 3, the DAFormer is a global and local transformer, which consists of a multi-heads cross attention and a group feed-forward network.

**Cross Attention.** The multi-heads cross-attention (MHCA) is used to learn global displacement features between $P_i$ and $P_{i-1}$. Denote two inputs with different scales as $X$ and $X'$. We can express the MHCA as:

$$q = \phi(X), k = \beta(X'), v = \psi(X'),$$
$$A = softmax(qk^T / \sqrt{C/h}), \tag{6}$$
$$MHCA(X, X') = \alpha(A \odot v),$$

where $\phi$, $\beta$, and $\psi$ are linear functions, $\alpha$ is a projection function to align the dimension. $C$ and $h$ are the embedding dimension and number of heads, respectively. $\odot$ is matrix multiplication.

**Group Feed-forward Network.** Different from the vanilla Transformer proposed by CrossVit (Chen et al., 2021), we employ a group feed-forward network (FFN) to introduce non-linear operation. Compared with regular FFN based on MLP, group FFN can aggregate the regional information by the grouping operation. Specifically, we first leverage a linear function to map the input channel into the hidden space, which is able to reduce the computing complexity. Next, we group the neighbors by the k-nearest neighbors (KNN) algorithm in the geometric space. Then, we use the relation between the center point and its neighbors to learn the displacement features. Finally, a back projection function is used to align the dimensions for the following residual connection. Note that, position encoding is not necessary for our DAFormer because we use group FFN to update the features. The output of the Group FFN can be formulated as:

$$\mathcal{G}(F) = \frac{1}{G} \sum_{j=1}^{G} \{\theta(\mathcal{M}\left([\phi(\mathbf{f}_i), \beta(\mathbf{f}_j)]\right))\} \tag{7}$$

where $\phi$, $\beta$ are linear function. $\theta$ is a back projection function. $\mathcal{M}$ is a mapping function with a series of MLP. The $G$ denotes the number of neighbors.

Benefiting from the cross attention and group FFN, our DAFormer can learn global and local displacements change to help following point shuffle operation to generate better displacements. The output of DAFormer can be designed as follows:

$$\begin{aligned} F' &= F_i + MHCA\left(LN(F_i), LN(F_{i-1})\right), \\ F_{out} &= F' + \mathcal{G}(LN(F')) \end{aligned} \tag{8}$$

where $\mathcal{G}$ is a group feed-forward network. $LN$ is the layer normalization (Ba et al., 2016).

## 3.4 Loss Function and Evaluation Metrics

We use the Chamfer Distance (CD) and F1-score to evaluate the quality of the reconstructed point clouds.

CD calculates the average closest point distances between the output $\mathbf{X}$ and the ground truth $\mathbf{Y}$, which can be defined as follows:

$$CD\left(X, Y\right) = \frac{1}{|X|} \sum_{x \in X} \min_{y \in Y} \|x - y\|_2 + \frac{1}{|Y|} \sum_{y \in Y} \min_{x \in X} \|y - x\|_2. \tag{9}$$

For an end-to-end training on our GANet, we design the total loss function, which is as follows:

$$\mathcal{L} = CD\left(P', Q\right) + \sum_{i=1}^{n} CD\left(P_i, Q\right), \tag{10}$$

where $P'$ and $P_i$ denote the coarse and fine output of DPA block, respectively. $Q$ is the ground truth. $n$ is the number of GAP blocks.

## 4 Experiments

**Implementation Details.** We build our network with PyTorch and CUDA. We train our models using an Adam optimizer (Kingma & Ba, 2014) with $\beta_1 = 0.9$ and $\beta_2 = 0.999$ on NVIDIA V100 16G GPU. The initial learning rate is set to $10^{-3}$ with a decay of 0.1 every 50 epochs, and the batch size is 32.

### 4.1 Results on MVP Dataset

**Dataset.** Pan et al. (2021) proposes a high-quality multi-view partial point cloud dataset (MVP) for the task of point cloud completion. It utilizes Poisson Disk Sampling (PDS) to generate the ground truth point clouds, and has multiple camera views (26 uniformly distributed camera poses on a unit sphere) and various categories. In addition, the MVP dataset provides different resolutions (i.e., 2048, 4096, 8192, and 16384) of ground truth for more accurate evaluation.

Table 1: Quantitative results on the MVP dataset (Pan et al., 2021) with different resolutions. "*" means additional data augmentation.

| Method | 2048 | | 4096 | | 8192 | | 16384 | |
|---|---|---|---|---|---|---|---|---|
| | CD (↓) | F1 (↑) | CD (↓) | F1 (↑) | CD (↓) | F1 (↑) | CD (↓) | F1 (↑) |
| PCN (Yuan et al., 2018) | 9.77 | 0.320 | 7.96 | 0.458 | 6.99 | 0.563 | 6.02 | 0.638 |
| TopNet (Tchapmi et al., 2019) | 10.11 | 0.308 | 8.20 | 0.440 | 7.00 | 0.533 | 6.36 | 0.601 |
| MSN (Liu et al., 2020) | 7.90 | 0.432 | 6.17 | 0.585 | 5.42 | 0.659 | 4.90 | 0.710 |
| CRN (Wang et al., 2020a) | 7.25 | 0.434 | 5.83 | 0.569 | 4.90 | 0.680 | 4.30 | 0.740 |
| VRCNet (Pan et al., 2021) | 5.96 | 0.499 | 4.70 | 0.636 | 3.64 | 0.727 | 3.12 | 0.791 |
| SnowflakeNet (Xiang et al., 2021a) | 5.76 | 0.513 | 4.42 | 0.671 | 3.50 | 0.746 | 2.74 | 0.800 |
| PoinTr (Yu et al., 2021) | - | - | 5.18 | 0.606 | 3.94 | 0.724 | 3.08 | 0.767 |
| PDR paradigm (Lyu et al., 2021) * | 5.66 | 0.499 | 4.26 | 0.649 | 3.35 | 0.754 | 2.61 | 0.817 |
| **GANet (ours)** | **4.99** | **0.527** | **3.81** | **0.679** | **2.87** | **0.776** | **2.28** | **0.828** |

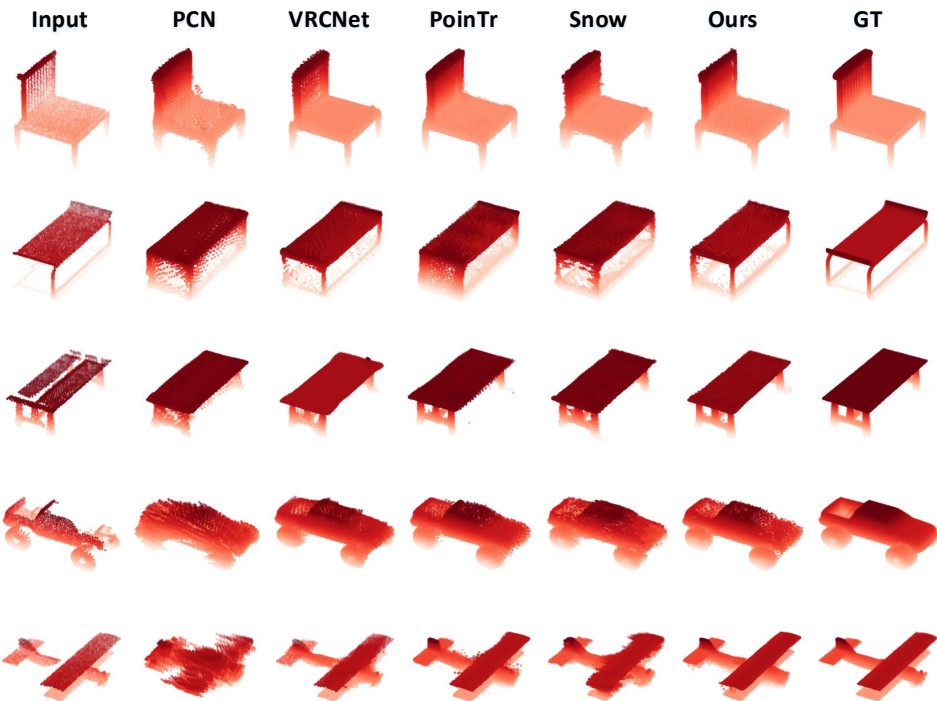

Figure 4: Visualization on MVP (Pan et al., 2021). From left to right, each column of images is incomplete point clouds (Input), the results of PCN (Yuan et al., 2018), PoinTr (Yu et al., 2021), SnowflakeNet (Xiang et al., 2021a), VRCNet (Pan et al., 2021), our GANet and ground truth (GT).

**Results.** To verify the effectiveness of our GANet, we evaluate our GANet on the MVP dataset and compare it with the previous methods in Table 1. Our method achieves state-of-the-art results on all of the resolutions in the metrics of CD and F1-score. Compared with state-of-the-art method PDR paradigm (Lyu et al., 2021), we reduce the 11.8% CD and improve 5.6% F1-score in the 2048 resolution.

Moreover, we visualize the reconstructed results in Figure 4. Compared with previous methods, our GANet is able to complete better and more detailed point clouds. Our GANet preserves the whole contour information of the input point clouds with the help of our graph-based local aggregators instead of the widely adopted PointNet-based feature extractors.

Table 2: Quantitative results on PCN dataset (Yuan et al., 2018) on CD ($\times 10^{-3}$). #Points of ground truth is 16384.

| Methods | Airplane | Cabinet | Car | Chair | Lamp | Sofa | Table | Watercraft | Avg ($\downarrow$) |
|---|---|---|---|---|---|---|---|---|---|
| FoldingNet (Yang et al., 2018) | 9.49 | 15.80 | 12.61 | 15.55 | 16.41 | 15.97 | 13.65 | 14.99 | 14.31 |
| AtlasNet (Groueix et al., 2018) | 6.37 | 11.94 | 10.10 | 12.06 | 12.37 | 12.99 | 10.33 | 10.61 | 10.85 |
| PCN (Yuan et al., 2018) | 5.50 | 22.70 | 10.63 | 8.70 | 11.00 | 11.34 | 11.68 | 8.59 | 9.64 |
| TopNet (Tchapmi et al., 2019) | 7.61 | 13.31 | 10.90 | 13.82 | 14.44 | 14.78 | 11.22 | 11.12 | 12.15 |
| CRN (Wang et al., 2020a) | 4.79 | 9.97 | 8.31 | 9.49 | 8.94 | 10.69 | 7.81 | 8.05 | 8.51 |
| GRNet (Xu et al., 2020b) | 6.45 | 10.37 | 9.45 | 9.41 | 7.96 | 10.51 | 8.44 | 8.04 | 8.83 |
| PMPNet (Wen et al., 2021b) | 5.65 | 11.24 | 9.64 | 9.51 | 6.95 | 10.83 | 8.72 | 7.25 | 8.73 |
| NSFA (Zhao et al., 2020) | 4.76 | 10.18 | 8.63 | 8.53 | 7.03 | 10.53 | 7.35 | 7.48 | 8.06 |
| PoinTr (Yu et al., 2021) | 4.75 | 10.47 | 8.68 | 9.39 | 7.75 | 10.93 | 7.78 | 7.29 | 8.38 |
| VE-PCN (Wang et al., 2021) | 4.80 | 9.85 | 9.26 | 8.90 | 8.68 | 9.83 | 7.30 | 7.93 | 8.32 |
| SnowflakeNet (Xiang et al., 2021a) | 4.29 | **9.16** | 8.08 | 7.89 | 6.07 | 9.23 | 6.55 | 6.40 | 7.21 |
| **GANet (ours)** | **3.98** | 9.21 | **7.86** | **7.43** | **5.53** | **8.91** | **6.35** | **6.14** | **6.92** |

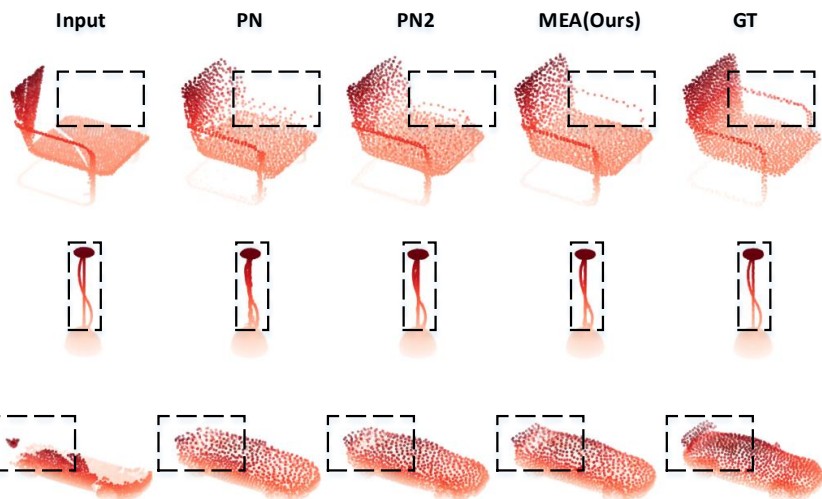

Figure 5: Visualization between different designed encoders. From left to right, each column of images is incomplete point clouds (Input), the results of encoders equipped with PointNet (PN) (Qi et al., 2017a), PointNet++ (PN2) (Qi et al., 2017b), and our MEA, and ground truth (GT).

## 4.2 RESULTS ON PCN DATASET.

**Dataset.** PCN (Yuan et al., 2018) uses synthetic CAD models from ShapeNet (Chang et al., 2015) to create a large-scale dataset containing numerous pairs of partial and complete point clouds. The 3D partial point clouds come from the back-projecting 2.5D depth maps. The ground truth with 16384 points is uniformly sampled from the model. This dataset includes 28974 CAD models for training and 1200 CAD models for test.

**Results.** We compare our GANet with recent most competitive methods on the PCN dataset (Yuan et al., 2018). Table 2 shows the quantitative results on the PCN dataset, from which we find that our proposed GANet performs best on the metric of CD.

## 5 ABLATION STUDY

In this section, we conduct detailed ablation studies to evaluate our designed components.

**The design of MEA.** We compare our MEA with PointNet-based encoder (Yuan et al., 2018) and PointNet2-based encoder (Wen et al., 2020). The quantitative results are shown

Table 3: Comparison between different encoders on CD ($\times 10^{-4}$) and F1-score. For the design of encoder, we have three options (PointNet (Qi et al., 2017a), PointNet++ (Qi et al., 2017b), and our MEA)

| PN | PN2 | MEA | CD ($\downarrow$) | F1-score ($\uparrow$) |
|---|---|---|---|---|
| ✓ | | | 5.25 | 0.526 |
| | ✓ | | 5.28 | 0.526 |
| | | ✓ | **4.99** | **0.527** |

Table 4: Ablation study of our DAFormer on CD ($\times 10^{-4}$) and F1-score. "Skip" means skip connection. "geo" is geometric neighbors, while "sem" is semantic neighbors.

| Skip | Block | neighbors | CD ($\downarrow$) | F1-score ($\uparrow$) |
|------|-------|-----------|-------------------|------------------------|
|  | MLP | - | 5.55 | 0.508 |
|  | Cross Transformer | - | 5.10 | 0.524 |
| ✓ | DAFormer | sem | 5.09 | 0.523 |
|  | **DAFormer(ours)** | geo | **4.99** | **0.527** |

in Table 3. The comparison demonstrates that our MEA achieves the best performance on CD and F1-score metrics. We visualize the results of the model with different encoders in Figure 5. The figure shows that the generated shape of our MEA is closer to ground truth in the whole contours compared with PointNet (PN) and PointNet++ (PN2). Both visualization and quantitative results prove that our graph-aware MEA improves the performance by exploiting the whole contour information.

**The design of LEA.** We ablate the design of LEA in our DPA. We compare our LEA with MLP in Table 5.

The results demonstrate that LEA is more powerful compared with vanilla MLP. We argue that the graph-based local aggregators help complete detailed point clouds by extracting geometric and structural features.

Table 5: Comparisons between MLP and LEA in our DPA block.

| Operation | CD ($\downarrow$) | F1-score ($\uparrow$) |
|-----------|-------------------|------------------------|
| MLP | 5.50 | 0.512 |
| LEA (ours) | **4.99** | **0.527** |

**The design of DAFormer.** We investigate the effectiveness of our DAFormer. All results testing on our GANet's architecture and the MVP dataset with 2048 resolutions. We first study the influence of skip-connection, the regular methods (Yuan et al., 2018; Tchapmi et al., 2019) without skip-connection usually use MLP to learn features. From the comparison, we find that without skip-connection, our model using MLP also achieves perfect performance, which still outperforms previous methods (Xiang et al., 2021a; Pan et al., 2021). Then, with the skip-connection, we compare the vanilla Cross Transformer (Chen et al., 2021) and our DAFormer. With the help of group FFN, our DAFormer gains better metrics in terms of CD and F1-score. Finally, we research the type of neighbor in the group FFN, including geometric neighbors and semantic neighbors. The experimented results demonstrate the geometric neighbor is better. We argue that the geometric neighbor constructs the feature fusion of geometric relation, which is able to generate more accurate displacements compared with the feature fusion of semantic relation.

## 6 COMPLEXITY ANALYSIS

In this section, we compare the efficiency between recent methods and our GANet on the MVP dataset with the resolution of 16384 points. As shown in Table 6, our GANet is much faster and brings obviously less computation compared with VRCNet (Pan et al., 2021), SnowflakeNet (Xiang et al., 2021a), and PoinTr (Yu et al., 2021). Moreover, Table 1 indicates GANet performs much better on CD and F1-score compared with these methods.

Table 6: Comparison between our GANet and the most recent methods on efficiency.

| Methods | FLOPs (G) | #Params (M) | Time (ms) |
|---------|-----------|-------------|-----------|
| VRCNet | 29.75 | 17.46 | 11.46 |
| Pointr | 11.14 | 42.55 | 10.12 |
| SnowflakeNet | 5.52 | 19.30 | 4.45 |
| **GANet** | 3.74 | 5.01 | 3.26 |

## 7 CONCLUSION

In this paper, we design a graph-aware network GANet for point cloud completion. Equipped with our proposed multi-scale edge aggregator, our GANet effectively learns from geometric global features from the preserved geometric structures of the partial point clouds. In addition, we propose the Displacement-Aware Point Augmentation (DPA) module to upsample the coarse output from the first

stage. With the graph-based feature extractor and Displacement-Aware Transformer (DAFormer), DPA generates accurate point displacements to upsample and refine point clouds. Extensive experiments on some benchmarks indicate that our GANet achieves state-of-the-art compared with previous methods.

## ETHICS STATEMENT

In our paper, we strictly adhere to ICLR ethical research standards and laws. All datasets we use are publicly available, and all relevant publications and source codes are appropriately cited.

## REPRODUCIBILITY STATEMENT

We adhere to ICLR reproducibility standards and ensure the reproducibility of our work in some ways as follows:

- We provide the codes of our main experiments in the supplementary material (code.zip), which includes the pretrained models and some demo samples.
- Detailed framework and more experiments are presented in the Appendix.

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

## A    Experiments details

In this section, we present our experimental setting in terms of upsampling ratios and our detailed experiment results on the MVP dataset.

**Upsampling ratios.** We set the size of coarse output to $512 \times 3$. In the different resolutions, the detailed upsampling ratios of three Displacements-Aware Point Augmentor (DPA) blocks are shown in following:

| Resolutions | Upsampling ratios |
| --- | --- |
| 2048 | 1,2,2 |
| 4096 | 1,2,4 |
| 8192 | 1,2,8 |
| 16384 | 1,2,16 |

**Detailed results on the MVP dataset.** We further present the detailed complete results on the MVP dataset (16384) in terms of F1-score. The results of every category are shown in following:

| Methods | airplane | cabinet | car | chair | lamp | sofa | table | watercraft | bed | bench | bookshelf | bus | guitar | motorbike | pistol | skateboard | Avg (↓) |
| --- | --- | --- | --- | --- | --- | --- | --- | --- | --- | --- | --- | --- | --- | --- | --- | --- | --- |
| PCN (Yuan et al., 2018) | 0.816 | 0.614 | 0.686 | 0.517 | 0.455 | 0.552 | 0.646 | 0.628 | 0.452 | 0.694 | 0.546 | 0.779 | 0.906 | 0.665 | 0.774 | 0.861 | 0.638 |
| TopNet (Tchapmi et al., 2019) | 0.789 | 0.621 | 0.612 | 0.443 | 0.387 | 0.506 | 0.639 | 0.609 | 0.405 | 0.680 | 0.524 | 0.766 | 0.868 | 0.619 | 0.726 | 0.837 | 0.601 |
| MSN (Liu et al., 2020) | 0.879 | 0.692 | 0.693 | 0.599 | 0.604 | 0.627 | 0.730 | 0.696 | 0.569 | 0.797 | 0.637 | 0.806 | 0.935 | 0.728 | 0.809 | 0.885 | 0.710 |
| CRN (Wang et al., 2020a) | 0.898 | 0.688 | 0.725 | 0.670 | 0.681 | 0.641 | 0.748 | 0.742 | 0.600 | 0.797 | 0.659 | 0.802 | 0.931 | 0.772 | 0.843 | 0.902 | 0.740 |
| GRNet (Xie et al., 2020) | 0.853 | 0.578 | 0.646 | 0.635 | 0.710 | 0.580 | 0.690 | 0.723 | 0.586 | 0.765 | 0.635 | 0.682 | 0.865 | 0.736 | 0.787 | 0.850 | 0.692 |
| NSFA (Zhang et al., 2020a) | 0.903 | 0.694 | 0.721 | 0.737 | 0.783 | 0.705 | 0.817 | 0.799 | 0.687 | 0.845 | 0.747 | 0.815 | 0.932 | 0.815 | 0.858 | 0.894 | 0.783 |
| VRCNet (Pan et al., 2021) | 0.928 | 0.721 | 0.756 | 0.743 | 0.789 | 0.696 | 0.813 | 0.800 | 0.674 | 0.863 | 0.755 | 0.832 | 0.960 | 0.834 | 0.887 | 0.930 | 0.796 |
| SnowflakeNet (Xiang et al., 2021a) | 0.928 | 0.729 | 0.731 | 0.750 | 0.806 | 0.722 | 0.815 | 0.801 | 0.701 | 0.866 | 0.756 | 0.834 | 0.966 | 0.815 | 0.877 | 0.924 | 0.800 |
| **GANet (Ours)** | **0.945** | **0.753** | **0.758** | **0.787** | **0.848** | **0.754** | **0.848** | **0.827** | **0.732** | **0.890** | **0.795** | **0.854** | **0.977** | **0.837** | **0.900** | **0.955** | **0.828** |

From the table, we can see that our GANet achieves state-of-the-art results in all categories.

## B    The design of edge aggregator.

We ablate the design of multi-scaling (the combination of $\mathbf{f}^{K_1}$, $\mathbf{f}^{K_2}$, $\mathbf{f}^{K_3}$ compared with $\mathbf{f}^{K_1}$ only) and multi-layer (three-layer MLP compared with single-layer MLP in $\mathcal{M}$). The results of testing on the MVP dataset with 2048 resolutions are shown in following.

| Scale | Layer | CD (↓) | F1-score (↑) |
| --- | --- | --- | --- |
| Single | Single | 5.19 | 0.525 |
| **Multi** | Single | 5.16 | 0.523 |
| Single | **Multi** | 5.08 | 0.524 |
| **Multi** | **Multi** | **4.99** | **0.527** |

From the comparison, multi-scale with multi-layer works the best on CD and F1-score.

## C    Results on the Real Scene

LiDAR scans can be very sparse, with some containing fewer than 10 points. KITTI (Geiger et al., 2013) is a real-world dataset that scanned point clouds using a Velodyne LiDAR. In order to compare the sensitivity and performance of models for the sparse point cloud in the LiDAR scans, we test the general models trained on the PCN ShapeNetCars. The visualized completion results are shown in Figure. 6. We compare our GANet with previous competitive methods, VRCNet (Pan et al., 2021), PoinTr (Yu et al., 2021) and SnowflakeNet (Xiang et al., 2021a). All the results show that our methods perform better in terms of details.

## D    More visualized results

In this section, we show more visualized complete results on the MVP dataset. From Figure 7, Figure 8 and Figure 9, we find that our method generate better shape with structure details compared with previous methods PCN (Yuan et al., 2018), VRCNet (Pan et al., 2021), PoinTr (Yu et al., 2021), and SnowflakeNet (Xiang et al., 2021a).

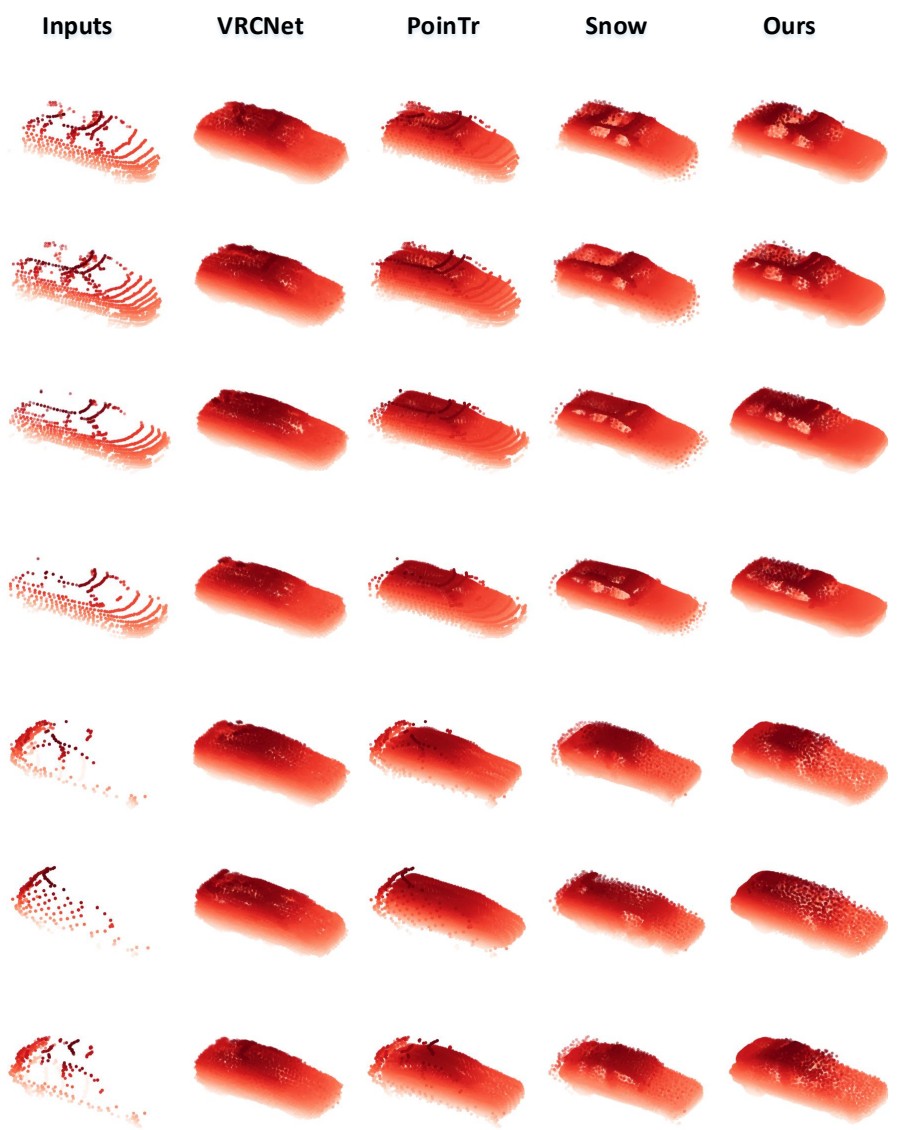

Figure 6: Visualized results on KITTI dataset.

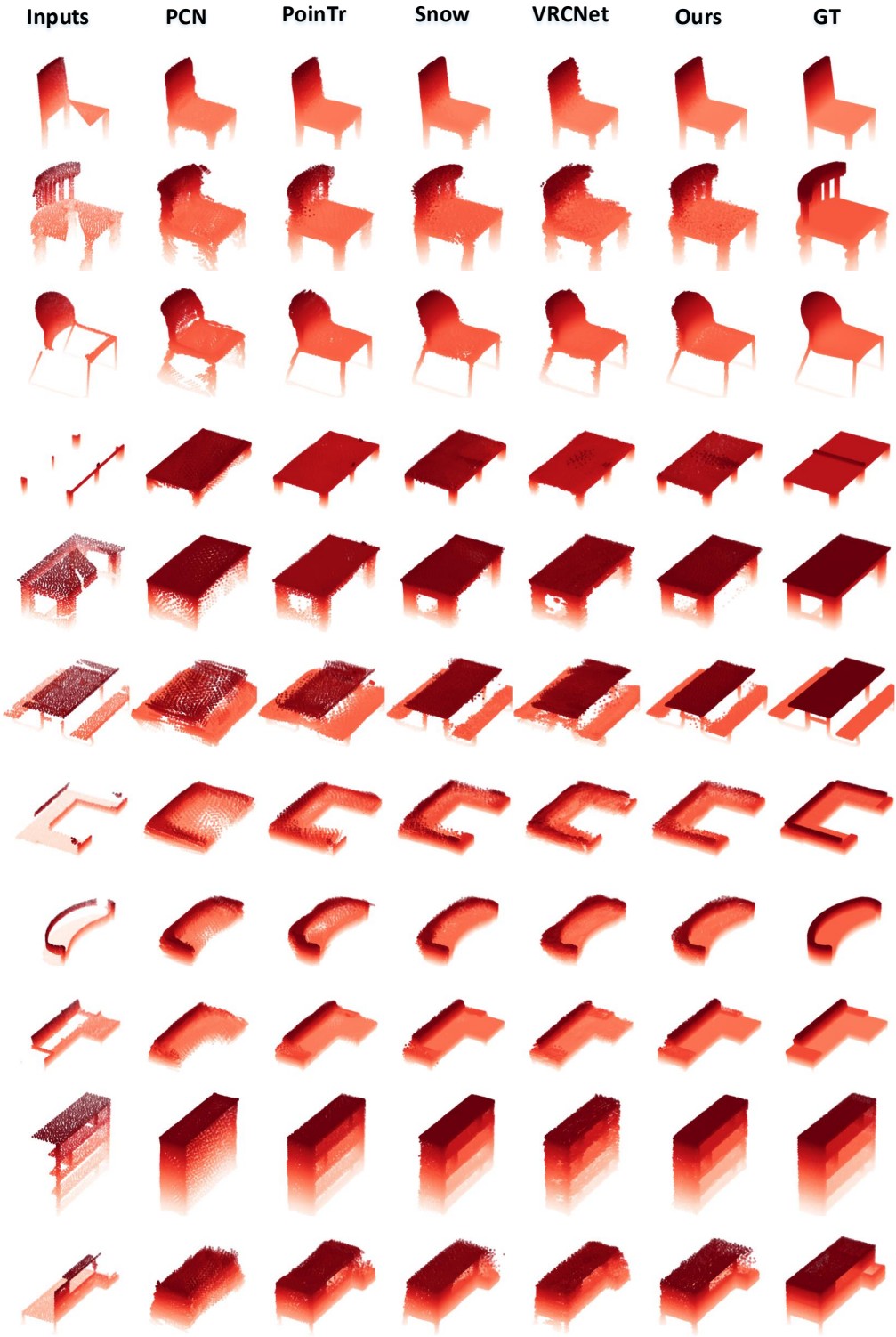

Figure 7: Visualized results on MVP dataset.

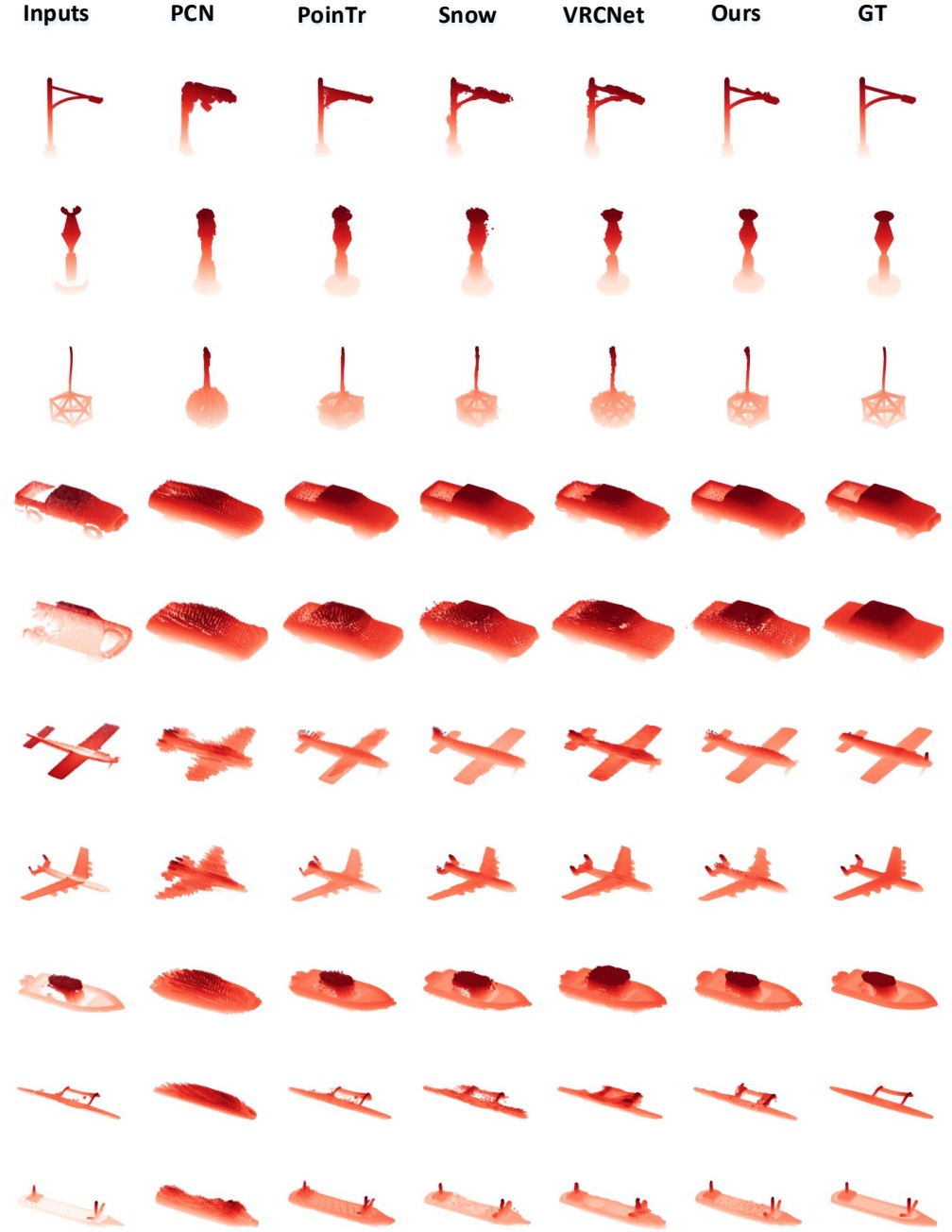

Figure 8: Visualized results on MVP dataset.

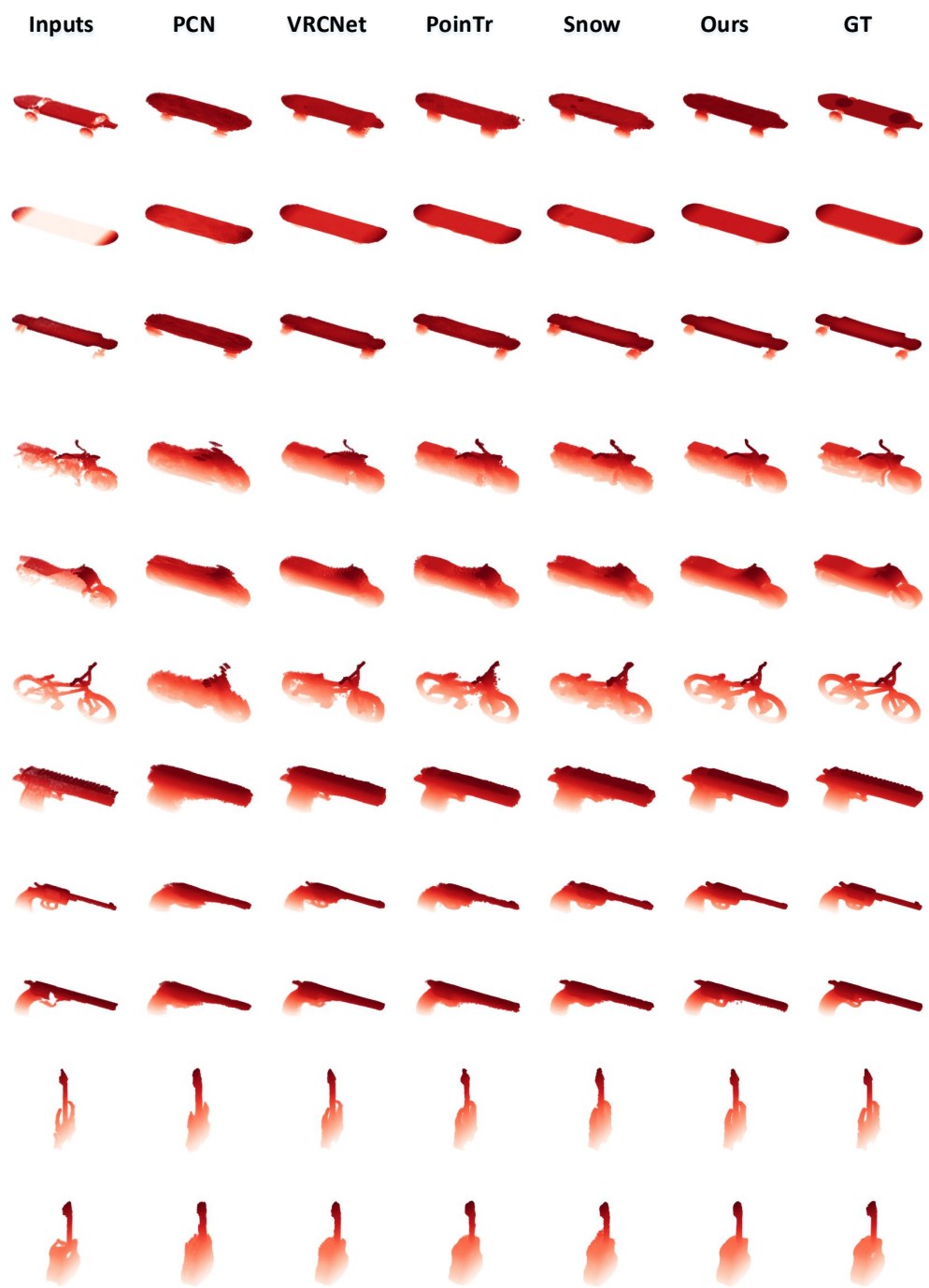

Figure 9: Visualized results on MVP dataset.

