# OpenReview forum: "GANet: Graph-Aware Network for Point Cloud Completion with Displacement-Aware Point Augmentor"
_ICLR.cc/2023/Conference — Submitted to ICLR 2023_

### Official Review · Reviewer_CpHT · 2022-10-23

**Confidence:** 5
**Correctness:** 4
**Technical Novelty And Significance:** 3
**Empirical Novelty And Significance:** 3
**Recommendation:** 10

**Clarity, Quality, Novelty And Reproducibility:**

The paper is well written, with proper use of the English language.
The authors explain their design choices in details and also provide code for reproducibility.
Many components of the work can be considered incremental but are well executed. The authors also propose a novel displacement aware transformer.

**Strength And Weaknesses:**

The paper is very well written. The proposed method is very interesting and shows very good performance both quantitatively and qualitatively. The authors describe well all their design choices. The paper is well evaluated and also includes ablation studies to support the design choices of the proposed method. The authors provide their code to allow reproducibility of their results. The method shows a consistent improvement over competing method. Overall, I think this paper is a solid submission and I recommend the paper to be accepted.

**Summary Of The Paper:**

The authors propose a graph aware deep neural network for point cloud completion. The design a novel displacement aware augmenter layer to effectively up sample and refine coarse point clouds. The authors demonstrate state of the art performance of their proposed method on PCN and MVP datasets.

**Summary Of The Review:**

The paper is well written. The proposed method demonstrates state of the art performance. The ideas are well explained and enough experiments and ablation studies have been conducted. I recommend the paper to be accepted.

---

> ### Author Response · Authors · 2022-11-19
> **Response to Reviewer CpHT**
>
> We sincerely thank you for the thoughtful feedback. For more details, please download our modified PDF, where words in blue indicate additions or revisions, and strikethrough words indicate deletions.
>
> As shown in our introduction, we propose our GANet motivated by the observation about the task of point cloud completion that most incomplete point clouds maintain roughly recognizable contours although difficult to discern as a whole. Firstly, we propose a Local Edge Aggregator (LEA) to extract local shape information based on the constructed dense graphs. We also introduce an attention mechanism to our LEA. Though simple, LEA achieves a better performance as shown in Table 5. Furthermore, to obtain a more expressive global representation, we further propose a Multi-scale Edge Aggregator (MEA) to capture different scales of local graph features. The ablation studies in Section B in Appendix and Table 3 show the effectiveness of this design. The quantitative results in Figure 5 further prove the effectiveness of our MEA. In addition, we propose DAFormer for point cloud refinement. It is capable of capturing long-range and local dependencies for accurate displacements. Equipped with the global attention mechanism and local group FFN, DAFormer boosts the performance of our architecture as shown in Table 4. Benefiting from the delicate design of our LEA, MEA, and DAFormer,  our GANet achieves state-of-the-art on the datasets of MVP and PCN. The visualization results in Figure 4 also prove its effectiveness. Simultaneously, our GANet is highly efficient compared with previous state-of-the-art methods in Table 6.
>
> Finally, we have provided our code for reproducibility, and we will release it publicly upon acceptance to help boost the community of point cloud completion.
>
> Thanks again for your positive response. Wish you a happy day!

---

### Official Review · Reviewer_97KL · 2022-10-24

**Confidence:** 4
**Correctness:** 4
**Technical Novelty And Significance:** 3
**Empirical Novelty And Significance:** 3
**Recommendation:** 6

**Clarity, Quality, Novelty And Reproducibility:**

Overall clarity of presentation is sufficient, however the use of various acronyms is a bit excessive and hinders general readability.

Quality of the written text and schematic visualizations of model components is good, visualization of point clouds is ok, but could be better.

Novelty is somewhat limited, since the model is mostly a combination of existing techniques.

The authors provided sufficient details and uploaded the code for their experiments, so reproducing results should be straightforward.

**Strength And Weaknesses:**

Strength:
* The model demonstrates state-of-the-art performance.
* Evaluation is done for two benchmarks and the chosen baselines are recent.
* Ablation studies are provided, showing how important different components are.
***
Weaknesses:
* Although complete model is very specifically engineered, overall novelty is limited, since the proposed techniques rely on existing prior ideas.

**Summary Of The Paper:**

This paper proposes a model for coarse-to-fine point cloud completion. The model first extracts global features from a partial input, then produces a coarse point cloud which is fed to several blocks in a sequential manner, each increasing the number of output points and modifying them. Overall, architecture heavily relies on the use of restricted local self-attention and multi-scale feature processing.

Architecture is evaluated on PCN and MVP datasets and compared to a large list of recent related models showing performance exceeding state of the art. Additionally, the authors provide several ablation studies, evaluating the performance of various model components; and compare their model in terms of memory footprint and computational efficiency.

**Summary Of The Review:**

Given overall quality of the submission including detailed introduction of the model, extensive experiments with two benchmarks and various ablation studies, and the resulting state-of-the-art performance, I think this paper is a good candidate for acceptance, even if the novelty of the approach is limited. If the authors will release their code, it will definitely be interesting to the community.

---

> ### Author Response · Authors · 2022-11-19
> **Response to reviewer 97KL**
>
> We sincerely thank you for the thoughtful feedback. For more details, please download our modified PDF, where words in blue indicate additions or revisions, and strikethrough words indicate deletions.
>
> As shown in our introduction, our novelty revolves around the observation about the task of point cloud completion that most incomplete point clouds maintain roughly recognizable contours although difficult to discern as a whole, and we leverage graph-based feature extraction to react to this observation. That is, to exploit the contour information for the task of point cloud completion. Firstly, we propose a Local Edge Aggregator (LEA) to extract local shape information based on the constructed dense graphs. We also introduce an attention mechanism to our LEA. Though simple, LEA achieves a better performance as shown in Table 5. Furthermore, to obtain a more expressive global representation, we further propose a Multi-scale Edge Aggregator (MEA) to capture different scales of local graph features. The ablation studies in Section B in Appendix and Table 3 show the effectiveness of this design.
>
> In addition, in the refinement stage, our proposed DAFormer is effective to capture long-range and local dependencies to generate better displacements. Based on the global attention mechanism and the local group FFN, our DAFormer further boosts the performance of our architecture as shown in the ablation study in Table 4.
>
> We attribute the excellent performance of our method on two popular benchmarks (MVP and PCN datasets) to the design of our MEA, LEA, and DAFormer. Besides, we have provided our code for reproducibility, and we will release it publicly upon acceptance to help boost the community of point cloud completion.
>
> Thank you again for your thoughtful response. Wish you a happy day!

---

### Official Review · Reviewer_NrG5 · 2022-10-24

**Confidence:** 4
**Correctness:** 2
**Technical Novelty And Significance:** 1
**Empirical Novelty And Significance:** Not applicable
**Recommendation:** 3

**Clarity, Quality, Novelty And Reproducibility:**

The presentation is unclear and difficult to understand.
The core component in the proposed method is not novel.

**Details Of Ethics Concerns:**

N.A.

**Strength And Weaknesses:**

Strength:
1) GANet achieves the state-of-the-art performance on PCN and MVP datasets.

Weakness:
1) Lack of clarity:

1.1) In Section 3.1, the input to Local Edge Aggregator (LEA) is defined as P’ (Nx3) and F’ (NxD). While in Figure 2-left, the input becomes P (NxC) and F is missing. The symbols used in Figure-2 left and right are also inconsistent.

1.2) In Section 3.1 Para. 1, “we use a different definition of edge”, what is this definition?

1.3) In Section 3.1, “we define K1, K2, K3 as 10, 20, N respectively”, what is N?

1.4) In Section 3.2, “the input of DPA is the point cloud Pi (Mx3)”, then what are the input point features?

1.5) Connection between Section 3.1 and 3.2 is missing. It is difficult for readers to understand the complete pipeline of the proposed method.

1.6) Symbols are also inconsistent between Section 3.1 and 3.2.


2) Lack of clear motivation:

2.1) The paper claims “Prior PointNet-based methods can hardly learn geometric information as they learn the global and local features from individual points” (Section 3.1) and “They (prior methods) usually employ MLPs-based feature extractor to learn the features of input, which may fail to exploit the geometric and structural features of the input” (Section 3.2) as the motivations of the proposed GANet. More in-depth analysis about the drawback of MLP-based feature extractor and the advantage of GANet is expected. Otherwise, the proposed method is not convincing. A simple quantitative comparison in Table 3 seems not enough.

2.2) It’s hard to find the relation between GANet and “graph-aware”. In its core graph-based Local Edge Aggregator (LEA), the operations shown in Equations (1) & (2) seem no difference from MLP-based paradigm.


3) Originality:
The core graph-based Local Edge Aggregator (LEA) is not novel. The paper does not clarify the definition of “edge” in this part, but:
--> If the “edge” is defined in Euclidean space of raw points, there is no difference from PointConv [1].
--> If the “edge” is defined in feature space, there is no difference from RandLA-Net [2].

[1] W. Wu et al, “PointConv: Deep Convolutional Networks on 3D Point Clouds”, CVPR 2019.
[2] Q. Hu et al, “RandLA-Net: Efficient Semantic Segmentation of Large-Scale Point Clouds”, CVPR 2020.



**Summary Of The Paper:**

This paper proposes a Graph-Aware Network (GANet) for point cloud completion. The network consists of 1) a graph-based Multi-scale Edge Aggregator (MEA) to extract global features from the input partial point clouds, and 2) a Displacements-aware Point Augmentor (DPA) to upsample and refine the point clouds. GANet achieves state-of-the-art performance on PCN and MVP datasets.

**Summary Of The Review:**

The problem this paper aims to address is not clear, and core component in the proposed method is not novel. Besides, the overall presentation can be significantly improved, especially in Section 3, due to the confusing organization and symbols.

---

> ### Author Response · Authors · 2022-11-19
> **Response to reviewer NrG5**
>
> We sincerely thank you for the thoughtful feedback. For more details, please download our modified PDF, where words in blue indicate additions or revisions, and strikethrough words indicate deletions.
>
> 1. Lack of clarity:
>
> We apologize for the typos or unclear explanations in our paper that may have affected your reading experience. To improve the readability of our paper, we have edited it based on your comments and will further proofread it.
>
> * 1-1. We have edited Section 3.1 and Figure 2 for symbol consistency. We further check the whole methodology part to guarantee the symbol consistency. For more details, please refer to Figure 2 and Section 3.1.
>
> * 1-2. This seems to be a typo we forgot to delete from a previous implementation, and we have deleted it in our modified paper. In this paper, we use the subtraction operation between features of the central point and its neighbor node as the definition of edge. For more details, please refer to the modified Section 3.1.
>
> * 1-3. "N" is short for "None", which indicates all points as neighbors. To avoid such a confusing term, we have added a detailed definition of "None" in Section 3.1.
>
> * 1-4. As shown in Section 3.2, the input of DPA contains two parts: a set of coordinates $P_{i} \subseteq \mathbb{R}^{N_i \times 3}$ output from the previous stage, and a set of features $F_{i-1}'\subseteq \mathbb{R}^{N_{i-1} \times D}$ from the previous stage after one LEA and one MLP.
>
> * 1-5. Our GANet is a coarse-to-fine method. In Section 3.1, we introduce Multi-scale Edge Aggregation to extract the global features of a coarse point cloud. Afterwards, in Section 3.2, we propose Displacement-aware Point Augmentation to refine the coarse point cloud. The overview of Section 3 further shows the connection between Section 3.1 and Section 3.2.
>
> * 1-6. We have edited Section 3.1 and 3.2 for symbol consistency. For more details, please refer to the two sections.
>
> 2. Lack of clear motivation
>
> * 2-1. Our main motivation is to reconstruct a complete shape by effectively extracting the contour information of a partial point cloud. As mentioned in our introduction, although difficult to discern as a whole, most incomplete point clouds maintain roughly recognizable contours. This observation motivates us to propose Graph-Aware Network (GANet), a novel graph-based network for shape completion. Compared with previous MLP-based approaches, which rely heavily on inductive learning and may neglect shape awareness as mentioned in [2], graph-based methods can more effectively extract contour information due to the hints of geometric relation. The ablation studies of our proposed Multi-Scale Edge Aggregator (MEA) and Local Edge Aggregator (LEA) indicate the effectiveness of our GANet in Table 3 and Table 5, respectively. Moreover, as shown in Figure 5 and Table 4, our proposed DAFormer, a block based on fully connected graphs, can further improve the performance of our GANet. Thanks to the design of our MEA, LEA, and DAFormer, our GANet achieves state-of-the-art on the popular benchmarks (MVP and PCN dataset) as shown in Table 1 and Table 2.
>
> * 2-2 Inspired by Dynamic Graph CNN in [1], we introduce pairwise Euclidean distances after constructing a dense graph of a group of points to our LEA. However, different from [1], our LEA leverages the combination of the attention mechanism and the constructed dense graph to extract local features. Though simple, this design further improves the performance of our method as shown in Table 5. Thus, we believe that the attention mechanism can help our architecture to be aware of dense graphs, so we named the architecture Graph-aware Network.
>
> [1] Wang, Yue, et al. "Dynamic graph cnn for learning on point clouds." Acm Transactions On Graphics (tog) 38.5 (2019): 1-12.\
> [2] Liu, Yongcheng, et al. "Relation-shape convolutional neural network for point cloud analysis." Proceedings of the IEEE/CVF Conference on Computer Vision and Pattern Recognition. 2019.

---

> > ### Author Response · Authors · 2022-11-19
> > **Response to reviewer NrG5 Part II**
> >
> > 3. Originality
> >
> > As shown in our introduction, our novelty revolves around the observation about the task of point cloud completion that most incomplete point clouds maintain roughly recognizable contours although difficult to discern as a whole, and we leverage graph-based feature extraction to react to this observation. That is, to exploit the contour information for the task of point cloud completion. For more expressive global representation, we further propose a Multi-scale Edge Aggregator (MEA) to capture different scales of local features. The ablation studies in Section B in Appendix and Table 3 show the effectiveness of this design. Besides, our proposed DAFormer based on fully connected graphs, is efficient to capture long-range and local dependencies for better refinement. It further boosts the performance of our architecture as shown in Table 4.
> >
> > Moreover, we apply feature subtraction to our Local Edge Aggregator (LEA) inspired by DGCNN [1]. However, it does not mean that our LEA is the same as RandLA-Net. The design of our LEA is based on the attention mechanism, for which we compute weights of an edge feature through a mapping function followed by a softmax function, while RandLA-Net adopts both coordinates and point features for local spatial encoding followed an attentive pooling. In addition, the attentive pooling is different from us -- attentive pooling computes weights based on the features themselves, but in our LEA, the weights and the features to be weighted are completely different in value or source.
> >
> > We attribute the excellent performance of our method on two popular benchmarks (MVP and PCN datasets) to the design of our MEA, LEA, and DAFormer. Besides, we have provided our code for reproducibility, and we will release it publicly upon acceptance to help boost the community of point cloud completion.
> >
> > Thank you again for your careful response. Wish you a happy day!

---

### Decision · Program_Chairs · 2023-01-20

**Decision:**

Reject

**Justification For Why Not Higher Score:**

This paper receives 1x reject, not good enough, 1x marginally above the acceptance threshold and 1x strong accept, should be highlighted at the conference. But the reviews that give accepts do not give very compelling reasons on why the proposed method is good. In contrast, the reject review pointed out several technical flaws of the paper.

**Justification For Why Not Lower Score:**

NA

**Metareview: Summary, Strengths And Weaknesses:**

This paper receives 1x reject, not good enough, 1x marginally above the acceptance threshold and 1x strong accept, should be highlighted at the conference.

The reject reviewer commented that the research problem that this paper aims to address is not clearly described, and core component in the proposed method is not novel. Furthermore, the overall presentation can be significantly improved, especially in Section 3, due to the confusing organization and symbols. More specifically, the reject reviewer pointed out the following flaws: (1) There’s a lack of clear motivation to develop the GANet to extract better features. Although the paper claims that earlier works such as PointCNN, KPConv, PointConv, DGCNN, RandLA-Net, Point Transformer, SparseConv, etc, cannot extract good local and geometric features, there's no support to their claim. (2) The motivation of learning “recognizable contours” is not convincing at all. The paper claims that the proposed GANet can better learn contours, but there's no more mention of contours in the design of the framework. (3) The proposed Displacement-aware Point Augmentation (DPA) block does not have a clear technique motivation. This component is somewhat new over existing works, but clear motivation, explanation, highlight, and extensive evaluations are missing. (4) The novelty of the proposed components is very limited. The paper lacks of clear motivations to design the feature extractor GANet, to learn contours, to design DPA module.

Although the marginally above the acceptance threshold reviewer feels that the paper has detailed introduction of the model, extensive experiments with two benchmarks and various ablation studies, and the resulting state-of-the-art performance, the novelty of the approach is limited.

The metareviewer carefully reads the reviews from all reviewers and find the both the marginally above the acceptance threshold and strong accept reviews do not give very compelling reasons on why the proposed method is good. In contrast, the reject review pointed out several technical flaws of the paper. Furthermore, the marginally above the acceptance threshold review also agrees that the proposed method is not novel.